# Old and New Definitions of Acute Respiratory Distress Syndrome (ARDS): An Overview of Practical Considerations and Clinical Implications

**DOI:** 10.3390/diagnostics15151930

**Published:** 2025-07-31

**Authors:** Cesare Biuzzi, Elena Modica, Noemi De Filippis, Daria Pizzirani, Benedetta Galgani, Agnese Di Chiaro, Daniele Marianello, Federico Franchi, Fabio Silvio Taccone, Sabino Scolletta

**Affiliations:** 1Department of Medical Science, Surgery and Neurosciences, Urgency-Emergency Anesthesia and Intensive Care Unit, University Hospital of Siena, 53100 Siena, Italy; cesare.biuzzi@ao-siena.toscana.it (C.B.); elena.modica@dbm.unisi.it (E.M.); noemi.defilippis@student.unisi.it (N.D.F.); daria.pizzirani@ao-siena.toscana.it (D.P.); benedetta.galgani@ao-siena.toscana.it (B.G.); agnese.dichiaro@ao-siena.toscana.it (A.D.C.); 2Department of Medical Science, Surgery and Neurosciences, Cardiothoracic and Vascular Anesthesia and Intensive Care Unit, University Hospital of Siena, 53100 Siena, Italy; d.marianello83@gmail.com (D.M.); federico.franchi@dbm.unisi.it (F.F.); 3Department of Intensive Care, Hôpital Universitaire de Bruxelles (HUB), Université Libre de Bruxelles (ULB), 1070 Brussels, Belgium; fabio.taccone@ulb.be

**Keywords:** ARDS, invasive and non-invasive mechanical ventilation, high-flow nasal cannula (HFNC), pneumonia

## Abstract

Lower respiratory tract infections remain a leading cause of morbidity and mortality among Intensive Care Unit patients, with severe cases often progressing to acute respiratory distress syndrome (ARDS). This life-threatening syndrome results from alveolar–capillary membrane injury, causing refractory hypoxemia and respiratory failure. Early detection and management are critical to treat the underlying cause, provide protective lung ventilation, and, eventually, improve patient outcomes. The 2012 Berlin definition standardized ARDS diagnosis but excluded patients on non-invasive ventilation (NIV) or high-flow nasal cannula (HFNC) modalities, which are increasingly used, especially after the COVID-19 pandemic. By excluding these patients, diagnostic delays can occur, risking the progression of lung injury despite ongoing support. Indeed, sustained, vigorous respiratory efforts under non-invasive modalities carry significant potential for patient self-inflicted lung injury (P-SILI), underscoring the need to broaden diagnostic criteria to encompass these increasingly common therapies. Recent proposals expand ARDS criteria to include NIV and HFNCs, lung ultrasound, and the SpO_2_/FiO_2_ ratio adaptations designed to improve diagnosis in resource-limited settings lacking arterial blood gases or advanced imaging. However, broader criteria risk overdiagnosis and create challenges in distinguishing ARDS from other causes of acute hypoxemic failure. Furthermore, inter-observer variability in imaging interpretation and inconsistencies in oxygenation assessment, particularly when relying on non-invasive measurements, may compromise diagnostic reliability. To overcome these limitations, a more nuanced diagnostic framework is needed—one that incorporates individualized therapeutic strategies, emphasizes lung-protective ventilation, and integrates advanced physiological or biomarker-based indicators like IL-6, IL-8, and IFN-γ, which are associated with worse outcomes. Such an approach has the potential to improve patient stratification, enable more targeted interventions, and ultimately support the design and conduct of more effective interventional studies.

## 1. Introduction

Respiratory complications are among the most significant clinical challenges in critically ill patients admitted to Intensive Care Units (ICUs). Among these, lower respiratory tract infections (LRTIs), in particular pneumonia, play a major role, as they frequently coexist with other critical conditions and contribute to worse outcomes [1,2]. The need for respiratory support is one of the primary reasons for ICU admission in these patients, with treatment options ranging from low- and high-flow oxygen therapy to non-invasive ventilation (NIV) and invasive mechanical ventilation [3]. The COVID-19 pandemic notably increased the adoption of HFNCs and NIV, driven in part by the limited availability of invasive mechanical ventilation [4]. Given the potential severity of pneumonia, timely diagnosis and appropriate treatment are crucial for improving clinical outcomes. For these reasons, these patients require comprehensive monitoring, which includes imaging studies such as lung ultrasonography (LUS), chest X-rays, and CT-scans, as well as the continuous assessment of respiratory drive, respiratory effort, and lung-distending pressures in order to prevent ventilator-induced lung injury (VILI) and patient self-inflicted lung injury (P-SILI), particularly in those receiving NIV and HFNC support, where monitoring can be less rigorous [5,6]. Microbiological sputum samples should be collected to ensure targeted antibiotic therapy against the responsible pathogen, and respiratory function must be assessed using parameters such blood gas analyses and the PaO_2_/FiO_2_ ratio via frequent arterial blood gas analysis [7]. In some cases, LRTIs and pneumonia can progress to acute respiratory distress syndrome (ARDS), a severe inflammatory condition leading to hypoxemia and respiratory failure, often necessitating orotracheal intubation and mechanical ventilation [8]. The Berlin ARDS criteria, which are actually used to define ARDS, serve as a key diagnostic tool assisting clinicians in defining this syndrome [9]; however, many patients with diffuse acute lung injury who receive HFNCs or NIV do not meet the Berlin definition for ARDS, suggesting that its incidence may be underdiagnosed. Indeed, one recent international observational study involving several Intensive Care Units across 50 countries indicated that ARDS remains underrecognized, with clinician recognition rates as low as 60%, particularly for mild cases; this phenomenon is associated with the insufficient application of recommended therapeutic measures (such as low tidal volume ventilation, the appropriate use of positive end-expiratory pressure, etc.), which contributes to persistently high mortality rates (approximately 40%) [8]. Therefore, a revised definition of ARDS has been proposed, potentially including patients treated with non-invasive ventilation, which could facilitate earlier diagnosis and more effective therapeutic interventions, improving overall patient outcomes [10].

In this brief clinical review, we aimed to examine the recent literature to provide an updated overview of the ARDS definition and explore potential approaches for refining both its classification and management.

## 2. Definition of ARDS

ARDS is a syndrome of respiratory failure with multiple etiologies that share common clinical–pathological characteristics, including the following: (a) increased permeability of the alveolo-capillary membrane, leading to inflammatory edema; (b) increased non-aerated lung tissue, resulting in higher lung elastance (lower compliance); and (c) increased venous admixture and dead space, which cause hypoxemia and hypercapnia [11]. First described in 1967 [12], the diagnostic criteria were revised in 1988 [13], and they have evolved significantly over the decades since 1994, when the American-European Consensus Conference (AECC) formalized diagnostic criteria [14] with the goal of refining and unifying the various existing definitions of ARDS and acute lung injury (ALI). In particular, the AECC established diagnostic criteria for ALI as a syndrome with acute onset, bilateral infiltrates on chest X-rays, arterial oxygenation (PaO_2_/FiO_2_ ≤ 300 mmHg), and no evidence of left atrial hypertension (wedge pressure ≤ 18 mmHg). ARDS was identified as a more severe form of ALI, with PaO_2_/FiO_2_ ≤ 200 mmHg [15]. Despite its utility, the AECC criteria faced criticism for lacking precision and clarity in several areas, such as variability in interpreting chest X-rays and the absence of severity stratification [16]. These limitations prompted the development of the Berlin definition in 2012 [9], whose key diagnostic criteria include an acute onset occurring within one week of a recognized clinical event or a significant worsening of respiratory symptoms and the presence of bilateral opacities on chest imaging (X-ray or CT) not fully attributable to other causes such as fluid overload, effusions, or lung collapse. Additionally, the diagnosis requires the exclusion of cardiogenic or hydrostatic pulmonary edema as the primary cause of respiratory failure and a PaO_2_/FiO_2_ ratio below 300 mmHg, with a minimum PEEP of 5 cmH_2_O or more [9,17]. This definition was primarily focused on patients undergoing invasive mechanical ventilation, effectively excluding those treated with non-invasive respiratory support. To overcome these limitations, several adaptations have been proposed. The Kigali modification (2016) substituted the PaO_2_/FiO_2_ ratio with the SpO_2_/FiO_2_ ratio, permitted lung ultrasound as an alternative to chest X-rays, and removed the PEEP requirement [18]. However, these adaptations introduce their own challenges. Lung ultrasound is highly operator-dependent, with image quality and interpretation varying by examiner expertise, while SpO_2_/FiO_2_ ratios can be skewed by motion artifacts, poor perfusion, and probe positioning [19,20]. In 2023, the European Society of Intensive Care Medicine (ESICM) updated its guidelines on ARDS, revising key aspects related to its definition, phenotyping, and respiratory support strategies (Table 1 and Table 2, Figure 1) [11]. Finally, a global consensus (2023–2024) advocated for expanding the ARDS definition to include patients treated with HFNCs (≥30 L/min) and NIV with equivalent PEEP, supported the use of SpO_2_/FiO_2_ for oxygenation assessment, and encouraged bedside diagnostic tools such as lung ultrasound (LUS). These updates aim to broaden diagnostic inclusivity, especially in low-resource settings, facilitate earlier recognition and intervention, and support a more personalized approach (e.g., combining imaging with biomarkers etc.) to patient management based on disease severity and the type of ventilatory support. However, challenges persist in avoiding overdiagnosis, maintaining diagnostic consistency across different clinical settings, and improving the accuracy and reliability of tools used to assess oxygenation [20].

## 3. Management of ARDS

The management of ARDS has advanced significantly in recent years. The cornerstone of ARDS treatment is supportive care, with a focus on lung-protective ventilation strategies, adjunctive therapies, and individualized approaches tailored to the patient’s clinical condition. The use of low tidal volumes (e.g., 6 mL/kg of predicted body weight) reduces the risk of ventilator-induced lung injury. Moreover, maintaining a plateau pressure below 30 cm H_2_O and a driving pressure below 15 cm H_2_O is equally important to prevent alveolar overdistension [21,22]. Positive end-expiratory pressure (PEEP) prevents alveolar collapse and improves oxygenation [10]. Furthermore, prone positioning enhances ventilation–perfusion matching and reduces alveolar collapse in dorsal regions; it has proven one of the most effective strategies for managing severe ARDS (PaO_2_/FiO_2_ below 150 mmHg), significantly reducing mortality in this high-risk group [8,23,24,25]. In selected refractory ARDS patients, the insertion of veno-venous extracorporeal membrane oxygenation (V-V ECMO) could improve gas exchange and enhance survival by enabling ultraprotective lung ventilation or lung rest [8]. The use of V-V ECMO was long considered only a rescue therapy for patients with severe ARDS. However, according to the latest ARDS guidelines from the European Society of Intensive Care Medicine (ESICM) of 2023, V-V ECMO is now recommended for selected patients with severe ARDS meeting specific eligibility criteria described by the EOLIA trial (2018): PaO_2_/FiO_2_ < 50 mmHg for over 3 h, <80 mmHg for over 6 h, or arterial pH < 7.25 with PaCO_2_ > 60 mmHg for over 6 h [26,27]. While the EOLIA trial did not demonstrate a significant reduction in 60-day mortality in ECMO-treated patients compared to those receiving only invasive mechanical ventilation, it highlighted an increased risk of bleeding or ischemic stroke. However, a meta-analysis by Munshi et al., 2019 [28], which included both the CESAR and EOLIA trials, demonstrated a significant reduction in 60-day mortality in patients treated with V-V ECMO [26,29,30].

## 4. NIV and HFNCs

NIV and HFNCs recently have emerged as valuable tools in managing ARDS, particularly in patients with mild-to-moderate hypoxemia, such as those in the early stages of respiratory failure or in resource-limited settings where invasive mechanical ventilation (MV) may not be immediately available.

### 4.1. Benefits

Non-invasive ventilation (NIV) and high-flow nasal cannulas (HFNCs) offer early respiratory support in patients with mild-to-moderate ARDS. These non-invasive strategies offer several advantages: in particular, the use of an HFNC became a critical respiratory support option during the COVID-19 pandemic, thanks to its ability to deliver heated, humidified oxygen at high flow rates. This reduces respiratory effort and improves oxygenation [26,30,31,32]. Moreover, HFNCs can reduce the need for invasive MV, thereby avoiding complications such as ventilator-induced lung injury (VILI), sedation-related risks, and ventilator-associated pneumonia [33]. Clinical trials have shown that HFNCs can lower tracheal intubation rates and improve survival compared to conventional oxygen therapy [34].

### 4.2. Risks

Despite these advantages, NIV and HFNCs carry significant hazards. Vigorous respiratory efforts can exacerbate lung injury through patient-self-inflicted lung injury (P-SILI)—a life-threatening condition in which excessive respiratory effort and forceful inspiration exacerbate lung damage, particularly in patients with pre-existing lung injury, severe ARDS, or a high inspiratory drive [35]. Furthermore, delayed intubation in patients who fail NIV or HFNCs has been associated with increased mortality, highlighting the importance of close monitoring and timely escalation to invasive MV when necessary [36,37].

The ESICM guidelines emphasize personalized treatment approaches based on ARDS phenotypes and suggest lung-protective ventilation strategies, such as low tidal volume, appropriate PEEP titration, and prone positioning. Additionally, they discuss the role of NIV and HFNCs, acknowledging their benefits but also cautioning against delayed tracheal intubation. These recommendations aim to optimize patient outcomes by integrating the latest evidence on MV, respiratory mechanics, and individualized treatment strategies [11].

## 5. New Criteria of ARDS

The increasing use of NIV and HFNCs in managing acute hypoxemic respiratory failure has contributed to the ongoing evolution of ARDS definitions, reflecting efforts to enhance diagnostic accuracy, improve clinical utility, and facilitate early intervention [16,36]. In 2023, a global consensus conference [10] with broad international representation suggested recommendations for updating the ARDS definition, identifying criteria that could be applied to all ARDS categories (risk factors and origin of edema, timing, chest imaging) and criteria that could be applied to specific ARDS categories [21].

### Rationale and Evidence

The Berlin definition, while a significant advancement in the diagnosis and classification of ARDS, has notable limitations. One critical point is the exclusion of patients receiving NIV respiratory support [38]. By requiring MV with a minimum PEEP of 5 cm H_2_O, the Berlin criteria leave out patients on HFNCs or NIV. These patients often show similar clinical and pathophysiological features to those who are intubated but are not included under the current ARDS definition. This gap may result in a failure to identify ARDS in its early stages for patients who do not require intubation, leading to delayed interventions and possibly worsening outcomes [39]. Additionally, the imaging criteria in the Berlin definition introduce significant subjectivity, as differences in interpretation between clinicians and imaging modalities—such as chest X-rays versus CT scans—can lead to inconsistencies in diagnosis. This variability affects the reproducibility and reliability of ARDS diagnosis, making its application in routine practice more challenging [40]. Another issue is the reliance on oxygenation thresholds, specifically the PaO_2_/FiO_2_ ratio, for identifying ARDS. This approach can lead to delayed diagnoses, as significant hypoxemia may not emerge until later in disease progression [10,41]. Moreover, recent evidence suggests that the PaO_2_/FiO_2_ ratio alone insufficiently captures the complexity of ARDS, particularly concerning ventilator-induced lung injury (VILI) [42]. Incorporating metrics such as the oxygenation factor (OF), which combines oxygenation data with mechanical variables like mean airway pressure (Paw) and positive end-expiratory pressure (PEEP), may offer a more comprehensive evaluation [43]. Finally, the Berlin definition’s dependence on arterial blood gas measurements and specific MV parameters limits its global applicability. In resource-limited settings, with limited access to advanced diagnostic tools, applying the current criteria is often not feasible [36]. The diagnostic criteria of ARDS and new definitions are reported in Table 3.

## 6. Advantages and Limits

### 6.1. Advantages and Practical Considerations

The new definition expands ARDS to include patients treated with HFNCs or NIV. This change acknowledges that many patients treated with non-invasive support referred to as “non-intubated ARDS” exhibit similar pathophysiological and clinical characteristics to those who are intubated [39]. Moreover, the inclusion of alternative oxygenation metrics, such as the SpO_2_/FiO_2_ ratio, facilitates the earlier recognition of ARDS [44]. This equivalence between SpO_2_/FiO_2_ and PaO_2_/FiO_2_ ratios was established through large-scale clinical validation studies demonstrating a strong linear correlation, leading to defined threshold values that reliably correspond to established PaO_2_/FiO_2_ diagnostic thresholds of 200 (SpO_2_/FiO_2_ ratio of 235) and 300 (SpO_2_/FiO_2_ ratio of 315) for ARDS and ALI, respectively [45,46]. These metrics are particularly useful in settings where arterial blood gas analysis may not be feasible, making the definition more accessible and practical for resource-limited environments [36]. A notable enhancement in the revised definition is its ability to stratify ARDS severity with greater accuracy. Adjustments to oxygenation thresholds and the introduction of PEEP-equivalent metrics for HFNC users enable a more accurate classification of severity in both intubated and non-intubated patients, allowing clinicians to personalize therapeutic strategies to meet individual patient needs and improve treatment effectiveness [10]. With the Kigali modification, bilateral B-lines or consolidations on lung-ultrasound (LUS) were allowed to fulfil the imaging criteria for ARDS. In comparison to the gold-standard computed tomography (CT) in high-resource settings, these criteria proved to be highly sensitive but with low to moderate specificity [47]. Recent international guidelines have increasingly recognized the potential value of lung ultrasound (LUS), as introduced by the Kigali modification, although it has not yet been formally incorporated into the standard global definition of ARDS [11]. In particular to maximize the benefits of LUS in ARDS a structured point-of-care ultrasound (POCUS) training is essential. Studies have shown that clinicians can achieve basic competency in lung ultrasound after as few as 20 supervised examinations, leading to significant improvements in image acquisition, interpretation accuracy, and time to diagnosis [48,49,50]. By standardizing POCUS training, scanning protocols, and interpretation criteria, teams can reduce inter-operator variability and integrate LUS more reliably into early ARDS management pathways. A useful development is the LUS-ARDS score—a data driven and externally validated method based on LUS-scores from both the left and right lungs—combined with the identification of an abnormal pleural line in the antero-lateral regions [49]. Although more complex than the Kigali modification, it shows higher accuracy in diagnosing and excluding ARDS [51]. Another critical advantage of the new definition is its improved global applicability, including in resource-limited settings. This adaptability ensures that ARDS can be diagnosed and effectively managed even in low-income countries where access to advanced diagnostics and MV is often restricted [36,52] (Table 4).

### 6.2. Limitations

The revised ARDS definition, while offering significant advancements, has several limitations that should be acknowledged. One prominent concern is the potential for overdiagnosis. By broadening the criteria to include non-intubated patients on HFNCs or NIV, there is a risk of misclassifying other causes of acute hypoxemic respiratory failure as ARDS. For example, non-invasive support techniques may suffer some technical problems like mask leaks or malpositioning that may result in uneven PEEP delivery, causing regional atelectasis and patchy aeration that appear as bilateral infiltrates. Similarly, modest fluid overload under insufficient positive pressure could generate interstitial B-lines indistinguishable from inflammatory edema. Under expanded HFNC-/NIV-based criteria, these patients could be misclassified as having ARDS, exposing them to unnecessary treatments and resource utilization, potentially overburdening healthcare systems and diluting the specificity of ARDS as a distinct clinical entity [39]. In addition, the application of the new definition is also problematic in certain specific patient populations, such as those receiving V-V ECMO. Many V-V ECMO patients are awake and spontaneously breathing and may even be in ambient air (FiO_2_ 21%). In such cases, traditional oxygenation-based diagnostic criteria such as PaO_2_/FiO_2_ or SpO_2_/FiO_2_ ratios can appear within normal limits despite the presence of severe lung pathology. As a result, these patients may not meet the formal ARDS criteria and could be erroneously excluded from diagnosis. As a result, the revised criteria may not fully address the diagnostic needs of ECMO-supported ARDS patients, leading to inconsistent classification [10].

The use of alternative oxygenation metrics, such as the SpO_2_/FiO_2_ ratio, is practical in resource-limited settings; however, they are inherently less accurate than arterial blood gas measurements. Additionally, recent studies highlight significant limitations of the SpO_2_/FiO_2_ ratio for ARDS severity classification, showing misclassification in about one-third of cases due to measurement inaccuracies and a high dependency on FiO_2_ settings [53]. Factors such as poor perfusion, skin pigmentation, patient movement, and device variability further limit the accuracy of SpO_2_-based measurements, emphasizing the need for careful interpretation and potential additional validation in clinical practice [54]. These inaccuracies may result in the overestimation or underestimation of ARDS severity, particularly in cases with borderline oxygenation status [36]. Additionally, the incorporation of LUS into ARDS diagnostics is valuable for bedside assessments, but on the other hand, it has some limitations, as its accuracy depends heavily on the clinician’s skill and experience and may be less effective in differentiating ARDS from other conditions with similar findings. In fact, LUS findings such as B-lines, pleural line abnormalities, or subpleural consolidations, though common in ARDS, are not pathognomonic and may also be present in conditions like cardiogenic pulmonary edema or interstitial lung diseases. This overlap can complicate the differential diagnosis, especially in the absence of standardized interpretation criteria [40,51,55] (Table 4).

## 7. Future Perspectives

The ongoing evolution of ARDS diagnostic criteria reflects the need for greater precision in identifying and managing this complex syndrome. However, several criticisms have emerged: variability and potential inaccuracies introduced by new indices and diagnostic tools; limited feasibility and lack of standardization of advanced physiological and imaging parameters; risks of overdiagnosis and consequent delays in appropriate interventions; and difficulties in implementing these approaches across diverse healthcare settings. Various authors have proposed refinements to improve diagnostic accuracy and to adapt the criteria more effectively for both clinical practice and research purposes [56].

One promising approach involves the use of PEEP-adjusted PaO_2_/FiO_2_ ratio: (P/FP) or SpO_2_/FiO_2_ (S/FP) ratios. These indices account for the level of positive end-expiratory pressure (PEEP) applied during ventilation, which plays a critical role in maintaining alveolar recruitment and optimizing oxygenation. P/FP is calculated by dividing the traditional PaO_2_/FiO_2_ ratio by the applied PEEP and then multiplying the result by a correction factor of 10. This correction factor was introduced to keep the P/FP ratio on a scale comparable to the traditional PaO_2_/FiO_2_ ratio, making it easier to interpret clinically. The same concept can be used for S/FP calculation. This formula adjusts for the contribution of PEEP to oxygenation, reflecting the interaction between alveolar recruitment and gas exchange efficiency [57]. However, several sources of inaccuracy may limit their reliability in fact they are vulnerable to practice-dependent PEEP settings and the arbitrary “×10” correction factor can reclassify patients without improving prediction. In addition, in non-invasive support (e.g., HFNCs), true PEEP is neither standardized nor directly measured, and SpO_2_ is prone to artifacts (poor perfusion, pigmentation), risking the misclassification of oxygenation status [57].

Some authors used the ROX index calculated as SpO_2_/FiO_2_ ratio/respiratory rate as an indicator of the severity of respiratory failure in patients with ARDS receiving non-invasive ventilation. A higher ROX index suggests better oxygenation efficiency relative to breathing effort, helping clinicians identify patients likely to benefit from continued non-invasive support versus those needing intubation. Despite its simplicity, the ROX index’s utility in ARDS is limited. It was originally developed and validated in pneumonia patients on high-flow nasal cannulas, so its performance in a broader ARDS population or in those receiving NIV may differ. Moreover, it is susceptible to SpO_2_ artifacts and the challenges of accurately measuring respiratory rate at the bedside [58,59].

For research purposes, several authors have emphasized the importance of integrating advanced physiological and imaging-based parameters into ARDS definitions [60,61]. These include measures of pulmonary vascular permeability, lung weight, and aeration. Pulmonary vascular permeability, which quantifies endothelial injury and capillary leak, is a direct indicator of the inflammatory and edematous processes underlying ARDS [62]. Ranieri and colleagues have argued that incorporating such measures could help distinguish ARDS from other causes of hypoxemic respiratory failure and offer a more refined characterization of disease severity [40]. However, pulmonary permeability indices often rely on invasive sampling or proprietary biomarkers that are not available in routine clinical practice, limiting their feasibility outside research centers [40]. Similarly, the assessment of lung weight and aeration through advanced imaging techniques, such as computed tomography (CT) or magnetic resonance imaging (MRI), provides valuable insights into the extent of alveolar flooding and consolidation. Nevertheless, quantitative CT and MRI provide high-resolution data but are costly, expose patients to radiation (in the case of CT), and require patient transport—an impractical option for unstable ICU patients. MRI also demands lengthy scan times and specialized expertise. In addition, inter-institutional variability in imaging protocols and post-processing algorithms further hampers comparability and may introduce measurement error. Until these hurdles are overcome, the integration of advanced permeability and imaging parameters into routine ARDS definitions will remain primarily a research endeavor rather than a universal clinical standard [63].

## 8. Conclusions

In summary, the development of ARDS diagnostic criteria and management strategies represent a significant effort to improve patient outcomes by adapting approaches to individual and contextual needs. Expanding the definition of ARDS to include non-invasive respiratory support tools, such as HFNCs and NIV, recognizes their crucial role in modern clinical practice, particularly in resource-limited settings and during global health crises, such as the COVID-19 pandemic. While these advancements undoubtedly improve diagnostic inclusivity and accessibility, they also bring challenges, such as potential overdiagnosis and variability in diagnostic tools like SpO_2_/FiO_2_ ratios and lung ultrasound. To avoid unnecessary treatments, it is essential that each new modality undergoes rigorous validation to confirm that it truly enhances diagnostic accuracy and clinical outcomes. Future perspectives in ARDS research and management point towards integrating advanced physiological indices like biomarkers, imaging modalities, and personalized approaches to refine diagnosis and stratify disease severity.

## Figures and Tables

**Figure 1 diagnostics-15-01930-f001:**
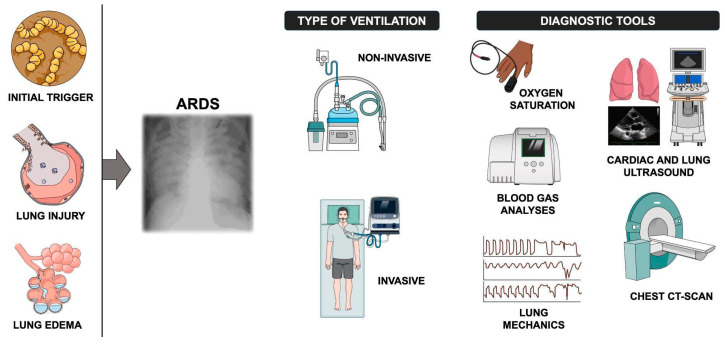
The figure shows the main steps leading to ARDS, the typical chest X-ray of the injured lungs, the ventilatory strategies (non-invasive and invasive), and the pivotal diagnostic tools used in daily clinical practice to assess the severity of the illness.

**Table 1 diagnostics-15-01930-t001:** Different definitions of ARDS for diagnosis.

DIAGNOSIS				
Criteria	Ashbaught 1967	AECC 1994	Berlin 2012	Kigali 2016
**Onset**	RF with tachypnea, lung stiffness	RF with tachypnea, lung stiffness	RF within 1 week not fully explained by cardiac function or volume overload	RF within 1 week not fully explained by cardiac function or volume overload
**Imaging**	Bilateral opacities on CRX	Bilateral opacities on CRX	Bilateral opacities on CRX or CT not fully explained by effusion, collapse or nodules	Bilateral opacities on CRX or US not fully explained by effusion, collapse or nodules
**Oxygenation**	Oxygenation impairment	Oxygenation impairment: ALI (P/F ≤ 300 mmHg) ARDS (P/F ≤ 200 mmHg)	Oxygenation impairment: Mild 200 < P/F ≤ 300 mmHg with PEEP ≥ 5 cmH_2_O Moderate 100 < P/F ≤ 200 mmHg with PEEP ≥ 5 cmH_2_O Severe P/F < 100 mmHg with PEEP ≥ 5 cmH_2_O	Oxygenation impairment: SpO_2_/FiO_2_ < 315; no PEEP requirement

RF: respiratory failure; CRX chest radiography; US ultrasound scan, CT: computed tomography scan; ALI: acute lung injury; PEEP: positive end-expiratory pressure; PaO_2_: partial pressure of arterial oxygen; FiO_2_: fraction of inspired oxygen; SpO_2_: oxygen saturation; P/F: PaO_2_/FiO_2_ ratio; ARDS: acute respiratory distress syndrome.

**Table 2 diagnostics-15-01930-t002:** Most important component of ARDS management, according to ESICM 2023 [11].

MANAGEMENT ESICM 2023
Low Tv ≤ 4–8 mL/kg PBW
Pplat ≤ 30 cmH_2_O, DP ≤ 15 cmH_2_O, Reduction Mechanical Power
Individualized PEEP titration, avoid lung recruitment maneuvers
Use of NIV or HFNCs to reduce risk of intubation
Prone Position (PaO_2_/FiO_2_ < 150, PEEP ≥ 5 cmH_2_O) and awake prone position
Use of ECMO VV in severe ARDS, avoid ECCO_2_R
Avoid continuous infusion of NMBA

Tv: tidal volume; PBW: predicted body weight; Pplat: plateau pressure; DP: driving pressure; PEEP: positive end-expiratory pressure; PaO_2_: partial pressure of arterial oxygen; FiO_2_: fraction of inspired oxygen; NIV: non-invasive ventilation; HFNC: high-flow nasal cannula; ECMO VV: veno-venous extracorporeal membrane oxygenation; ECCO_2_R: extracorporeal carbon dioxide removal; NMBA: neuro-muscular blocking agents. Note Table 2: Recommendations based on ESICM 2023 consensus [11]; see also supporting literature [12,13,14,15,16] for specific interventions.

**Table 3 diagnostics-15-01930-t003:** Diagnostic criteria of ARDS in the new definition and differences from the Berlin criteria.

NEW ARDS DEFINITION	BERLIN DEFINITION
** Criteria for ALL ARDS Categories **	
Risk factor and origin of edema: triggered by an acute predisposing risk factor, such as pneumonia, non-pulmonary infection, trauma, transfusion, aspiration, or shock. Timing: rapid development or progression of hypoxemic respiratory failure within one week of the suspected trigger or new/worsening respiratory symptoms. Chest imaging: bilateral opacities on CRX or CT-scan is not explained by effusions, atelectasis or nodules/masses. The new definition also considers ultrasonography as a diagnostic imaging method.	
** Criteria for SPECIFIC ARDS Categories ** ** NEW ARDS DEFINITION **	** Criteria for SPECIFIC ARDS Categories ** ** BERLIN DEFINITION **
Non-intubated ARDS ○ **P/F** ≤ 300 mmHg ○ **SpO_2_/FiO_2_** ≤ 315 (if SpO_2_ ≤ 97%) on HFNCs with flow ≥ 30 L/min or NIV/CPAP with at least 5 cmH_2_O PEEP Intubated ARDS ○ Mild 200 < **P**/**F** ≤ 300 mmHg with PEEP ≥ 5 cmH_2_O 235 < **SpO_2_**/**FiO_2_** ≤ 315 (if SpO_2_ ≤ 97%) ○ Moderate 100 < **P/F** ≤ 200 mmHg with PEEP ≥ 5 cmH_2_O 148 < **SpO_2_**/**FiO_2_** ≤ 235 (if SpO_2_ ≤ 97%) ○ Severe **P**/**F** < 100 mmHg with PEEP ≥ 5 cmH_2_O **SpO_2_**/**FiO_2_** ≤ 148 (if SpO_2_ ≤ 97%) Modified definition for resource variable settings ○ **SpO_2_**/**FiO_2_** ≤ 315 (if SpO_2_ ≤ 97%). Minimum PEEP not required	Mild 200 < **P**/**F** ≤ 300 mmHg with PEEP ≥ 5 cmH_2_O Moderate 100 < **P**/**F** ≤ 200 mmHg with PEEP ≥ 5 cmH_2_O Severe **P**/**F** < 100 mmHg with PEEP ≥ 5 cmH_2_O

PaO_2_: partial pressure of arterial oxygen; FiO_2_: fraction of inspired oxygen; SpO_2_: saturation of oxygen; P/F: PaO_2_/FiO_2_ ratio; PEEP: positive end-expiratory pressure; NIV: non-invasive ventilation; HFNC: high-flow nasal cannula; CPAP: continuous positive airway pressure.

**Table 4 diagnostics-15-01930-t004:** Practical advantages and limitations of new ARDS definition.

NEW ARDS DEFINITION			
New Definition	Advantages	Limitations	Clinical Implications
**Inclusion of HFNCs/NIV**	Expands recognition of “non intubated ARDS”	Potential for overdiagnosis	Closer monitoring needed to avoid delayed intubation
**SpO_2_/FiO_2_ for diagnosis**	Useful in low-resource settings	Affected by perfusion, skin pigmentation, device accuracy	May require arterial blood gas confirmation
**Use of lung ultrasound**	Portable, bedside diagnostic tool	Operator-dependent, lacks standardized criteria	Training and standardization are essential
**Applicability in resource-limited settings**	Does not require PEEP for diagnosis	Excludes ECMO patients	May help early diagnosis but could overdiagnose ARDS

HFNC: high-flow nasal cannula, NIV: non-invasive ventilation, SpO_2_/FiO_2_: saturation of oxygen/fraction of inspired oxygen.

## Data Availability

Not applicable.

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
