# Peer review of "Old and New Definitions of Acute Respiratory Distress Syndrome (ARDS): An Overview of Practical Considerations and Clinical Implications"

_diagnostics, 2025, doi:10.3390/diagnostics15151930_

Round 1
Reviewer 1 Report
Comments and Suggestions for Authors
The topic is interesting and quite well written. I have some comments:
1) Abstract. The paragraph covers multiple important points but could benefit from clearer segmentation. For example, separating the discussion of the Berlin definition, recent modifications, and associated challenges into distinct sections would improve readability and comprehension.
2) Abstract. Some terms, like "conduction of interventional trials," could be phrased more clearly as "conducting interventional trials" or "the design of interventional studies" for better clarity.
3) Introduction. The COVID-19 50 pandemic notably increased the adoption of HFNC and NIV, driven in part by the limited 51 availability of invasive mechanical ventilation [4]. Authors are kindly requested to emphasize the current concepts about these issues in the context of recent knowledge and the available literature. These articles should be quoted in the References list:
a- Respiratory Drive, Effort, and Lung-Distending Pressure during Transitioning from Controlled to Spontaneous Assisted Ventilation in Patients with ARDS: A Multicenter Prospective Cohort Study. J Clin Med. 2024;13(17):5227. doi:10.3390/jcm13175227; b- Different Methods to Improve the Monitoring of Noninvasive Respiratory Support of Patients with Severe Pneumonia/ARDS Due to COVID-19: An Update. J Clin Med. 2022;11(6):1704. doi:10.3390/jcm11061704
4) 2. Definition Of ARDS. The paragraphs offer a comprehensive overview but would benefit from sharper focus, critical engagement with the literature, and elaboration on the implications of recent changes in ARDS diagnosis and management.
5) 4.2. Limitations. I suggest to underline and clarify the most important limitations.
6) 5. Future Perspectives. I would like to highlight some criticisms: - variability and potential inaccuracies introduced by new indices and diagnostic tools.
- limited feasibility and standardization of advanced physiological and imaging parameters.
- risks of overdiagnosis and delayed appropriate interventions.
- difficulties in implementing these approaches in different healthcare settings.
6) 6. Conclusions. The main criticisms are the potential for overdiagnosis and variability in diagnostic tools, as well as the challenge of ensuring that the inclusion of new modalities truly improves diagnostic accuracy without leading to unnecessary treatments.
Author Response
We thank this Reviewer for his/her helpful suggestions and comments. We appreciated his/her feedback about our article very much. Thanks to this, we had the opportunity to significantly improve the paper.
C: Abstract. The paragraph covers multiple important points but could benefit from clearer segmentation. For example, separating the discussion of the Berlin definition, recent modifications, and associated challenges into distinct sections would improve readability and comprehension.
R: We thank the reviewer for the suggestion. As requested, we used a clearer segmentation of the Abstract. (Lines 16-43)
Low respiratory tract infections remain a leading cause of morbidity and mortality among Intensive Care Unit patients, with severe cases often progressing to acute respiratory distress syndrome (ARDS). This life-threatening syndrome results from alveolar–capillary membrane injury, causing refractory hypoxemia …
The 2012 Berlin definition standardized ARDS diagnosis but excluded patients on non-invasive ventilation (NIV) or high-flow nasal cannula (HFNC) modalities, which are increasingly used, especially after the COVID-19 pandemic. By excluding these patients, diagnostic delays can occur, risking progression of lung injury despite ongoing support. Indeed, sustained, vigorous respiratory efforts under non‑invasive modalities carry a significant potential for patient self‑inflicted lung injury (P‑SILI), underscoring the need to broaden diagnostic criteria to encompass these increasingly common therapies.
Recent proposals expand ARDS criteria to include NIV and HFNC, lung ultrasound, and the SpO₂/FiO₂ ratio adaptations designed to improve diagnosis in resource-limited settings lacking arterial blood gases or advanced imaging.
However, broader criteria risk overdiagnosis and create challenges in distinguishing ARDS from other causes of acute hypoxemic failure …
To overcome these limitations, a more nuanced diagnostic framework is needed, one that incorporates individualized therapeutic strategies, emphasizes lung-protective ventilation, and integrates advanced physiological or biomarker-based indicators like IL-6, IL-8, IFN-γ associated with worse outcomes.
Such an approach has the potential to improve patient stratification, enable more targeted interventions, and ultimately support the design and conduct of more effective interventional studies.
C: Abstract. Some terms, like "conduction of interventional trials," could be phrased more clearly as "conducting interventional trials" or "the design of interventional studies" for better clarity.
R: We agree with the Reviewer. We phrased clearly the text for better clarity. (Lines 16-43)
Such an approach has the potential to improve patient stratification, enable more targeted interventions, and ultimately support the design and conduct of more effective interventional studies.
C: Introduction. The COVID-19 pandemic notably increased the adoption of HFNC and NIV, driven in part by the limited availability of invasive mechanical ventilation [4]. Authors are kindly requested to emphasize the current concepts about these issues in the context of recent knowledge and the available literature. These articles should be quoted in the References list:
a- Respiratory Drive, Effort, and Lung-Distending Pressure during Transitioning from Controlled to Spontaneous Assisted Ventilation in Patients with ARDS: A Multicenter Prospective Cohort Study. J Clin Med. 2024;13(17):5227. doi:10.3390/jcm13175227; b- Different Methods to Improve the Monitoring of Noninvasive Respiratory Support of Patients with Severe Pneumonia/ARDS Due to COVID-19: An Update. J Clin Med. 2022;11(6):1704. doi:10.3390/jcm11061704
R: As requested, we emphasized the current concepts quoting the suggested references. We have also updated the text according with the suggestions of Reviewer 3. (Lines 46-84)
… For this reasons, these patients require comprehensive monitoring, which includes imaging studies such as lung ultrasonography (LUS), chest X-rays, CT-scans as well as continuous assessment of respiratory drive, respiratory effort, and lung-distending pressures in order to prevent Ventilator-Induced Lung Injury (VILI) and Patient Self-Inflicted Lung Injury (P-SILI) particularly in those receiving NIV and HFNC support where monitoring can be less rigorous [5,6]…
… Indeed, one recent international observational study involving several intensive care units across 50 countries indicates that ARDS remains underrecognized, with clinician recognition rates as low as 60%, particularly for mild cases; this phenomenon is associated with insufficient application of recommended therapeutic measures (such as low tidal volume ventilation, appropriate use of positive end-expiratory pressure etc., Table 2) which contribute to persistently high mortality rates (approximately 40%) [8]…
C: Definition Of ARDS. The paragraphs offer a comprehensive overview but would benefit from sharper focus, critical engagement with the literature, and elaboration on the implications of recent changes in ARDS diagnosis and management.
R: We thank the reviewer for this comment. We have tightened the text to emphasize the clinical and research implications of recent ARDS definition updates. We have also updated the text according with the suggestions of Reviewer 3. (Lines 85-128)
… First described in 1967 [12] his diagnostic criteria were revised in 1988 [13], and evolved significantly over the decades since 1994 when the American-European Consensus Conference (AECC) formalized diagnostic criteria [14] with the goal of refining and unifying the various existing definitions of ARDS and Acute Lung Injury (ALI). In particular AECC established diagnostic criteria for ALI as a syndrome with acute onset …
These limitations prompted the development of the Berlin definition in 2012 [9] whose key diagnostic criteria include an acute onset occurring within one week of a recognized clinical event or a significant worsening of respiratory symptoms; … Additionally, the diagnosis requires the exclusion of cardiogenic or hydrostatic pulmonary edema as the primary cause of respiratory failure and a PaO₂/FiO₂ ratio below 300 mmHg, with a minimum PEEP of 5 cm H₂O or more [9,17]. This definition was primarily focused on patients undergoing invasive mechanical ventilation, effectively excluding those treated with non-invasive respiratory support. To overcome these limitations, several adaptations have been proposed. The Kigali modification (2016) substituted the PaO₂/FiO₂ ratio with the SpO₂/FiO₂ ratio, permitted lung ultrasound as an alternative to chest X-rays, and removed the PEEP requirement [18]. However, these adaptations introduce their own challenges. Lung ultrasound is highly operator-dependent, with image quality and interpretation varying by examiner expertise, while SpO₂/FiO₂ ratios can be skewed by motion artifacts, poor perfusion, and probe positioning [19,20]. In 2023, the European Society of Intensive Care Medicine (ESICM) updated its guidelines on ARDS, revising key aspects related to its definition, phenotyping, and respiratory support strategies (Table 1 and 2, Figure 1) [11]. Finally, a global consensus (2023–2024) advocated for expanding the ARDS definition to include patients treated with HFNC (≥30 L/min) and NIV with equivalent PEEP, supported the use of SpO₂/FiO₂ for oxygenation assessment, and encouraged bedside diagnostic tools such as lung ultrasound (LUS). These updates aim to broaden diagnostic inclusivity, especially in low-resource settings, facilitate earlier recognition and intervention, and support a more personalized approach (e.g., combining imaging with biomarkers etc.) to patient management based on disease severity and the type of ventilatory support. However, challenges persist in avoiding overdiagnosis, maintaining diagnostic consistency across different clinical settings, and improving the accuracy and reliability of tools used to assess oxygenation [20].
C: Limitations. I suggest to underline and clarify the most important limitations.
R: We underlined and clarified the most important limitations. We have also updated the text including the changes requested by Reviewer 3. (Lines 252-332)
Advantages and Practical Considerations
… Recent international guidelines have increasingly recognized the potential value of lung ultrasound (LUS), as introduced by the Kigali modification, although it has not yet been formally incorporated into the standard global definition of ARDS [11]. In particular to maximize the benefits of LUS in ARDS a structured point-of-care ultrasound (POCUS) training is essential. Studies have shown that clinicians can achieve basic competency in lung ultrasound after as few as 20 supervised examinations, leading to significant improvements in image acquisition, interpretation accuracy, and time to diagnosis [47–49]. By standardizing POCUS training, scanning protocols, and interpretation criteria, teams can reduce inter-operator variability and integrate LUS more reliably into early ARDS management pathways. A useful development is the LUS-ARDS score, a data driven and externally validated method based on LUS-scores from both the left and right lungs, combined with the identification of an abnormal pleural line in the antero-lateral regions [50]. Although more complex than the Kigali modification, it shows higher accuracy in diagnosing and excluding ARDS [51]. …
… This adaptability ensures that ARDS can be diagnosed and effectively managed even in low-income countries where access to advanced diagnostics and MV is often restricted [35,52] (Table 4).
Limitations
… By broadening the criteria to include non-intubated patients on HFNC or NIV, there is a risk of misclassifying other causes of acute hypoxemic respiratory failure as ARDS. For example non‐invasive support techniques may suffer some technical problems like mask leaks or malpositioning that may result in uneven PEEP delivery causing regional atelectasis and patchy aeration that appear as bilateral infiltrates. Similarly, modest fluid overload under insufficient positive pressure could generate interstitial B-lines indistinguishable from inflammatory edema. Under expanded HFNC/NIV-based criteria, these patients could be misclassified as ARDS, exposing patients to unnecessary treatments and resource utilization, potentially overburdening healthcare systems and diluting the specificity of ARDS as a distinct clinical entity [38]. In addition, the application of the new definition is also problematic in certain specific patient populations, such as those receiving V-V ECMO. Many V-V ECMO patients are awake and spontaneously breathing and may even be in ambient air (FiO₂ 21%). In such cases, traditional oxygenation-based diagnostic criteria such as PaO₂/FiO₂ or SpO₂/FiO₂ ratios can appear within normal limits despite the presence of severe lung pathology. As a result, these patients may not meet the formal ARDS criteria and could be erroneously excluded from diagnosis. As a result, the revised criteria may not fully address the diagnostic needs of ECMO-supported ARDS patients, leading to inconsistent classification [10].
… Additionally, the incorporation of LUS into ARDS diagnostics is valuable for bedside assessments, but on the other hand has some limitations, as its accuracy depends heavily on the clinician’s skill and experience and may be less effective in differentiating ARDS from other conditions with similar findings. In fact LUS findings such as B-lines, pleural line abnormalities, or subpleural consolidations, though common in ARDS, are not pathognomonic and may also be present in conditions like cardiogenic pulmonary edema or interstitial lung diseases. This overlap can complicate the differential diagnosis, especially in the absence of standardized interpretation criteria [39,51,55] (Table 4).
C: Future Perspectives. I would like to highlight some criticisms:
- variability and potential inaccuracies introduced by new indices and diagnostic tools.
- limited feasibility and standardization of advanced physiological and imaging parameters.
- risks of overdiagnosis and delayed appropriate interventions.
- difficulties in implementing these approaches in different healthcare settings.
R: Thanks for this comment. We changed the text and also made additional changes according to the suggestions of Reviewer 3. (Lines 335-390)
The ongoing evolution of ARDS diagnostic criteria reflects the need for greater precision in identifying and managing this complex syndrome. However, several criticisms have emerged: variability and potential inaccuracies introduced by new indices and diagnostic tools; limited feasibility and lack of standardization of advanced physiological and imaging parameters; risks of overdiagnosis and consequent delays in appropriate interventions; and difficulties in implementing these approaches across diverse healthcare settings. Various authors have proposed refinements to improve diagnostic accuracy and to adapt the criteria more effectively for both clinical practice and research purposes [56].
… P/FP is calculated by dividing the traditional PaO₂/FiO₂ ratio by the applied PEEP and then multiplying the result by a correction factor of 10. This correction factor was introduced to keep the P/FP ratio on a scale comparable to the traditional PaO₂/FiO₂ ratio, making it easier to interpret clinically. The same concept can be used for S/FP calculation. This formula adjusts for the contribution of PEEP to oxygenation, reflecting the interaction between alveolar recruitment and gas exchange efficiency [57]. However, several sources of inaccuracy may limit their reliability in fact they are vulnerable to practice‐dependent PEEP settings and the arbitrary “×10” correction factor can reclassify patients without improving prediction. In addition, in noninvasive support (e.g., HFNC), true PEEP is neither standardized nor directly measured, and SpO₂ is prone to artifacts (poor perfusion, pigmentation), risking misclassification of oxygenation status [57].
Some authors used the ROX index calculated as SpO₂/FiO₂ ratio/respiratory rate as an indicator of the severity of respiratory failure in patients with ARDS receiving non-invasive ventilation. A higher ROX index suggests better oxygenation efficiency relative to breathing effort, helping clinicians identify patients likely to benefit from continued non-invasive support versus those needing intubation. Despite its simplicity, the ROX index’s utility in ARDS is limited. It was originally developed and validated in pneumonia patients on high-flow nasal cannula, so its performance in a broader ARDS population or in those receiving NIV may differ. Moreover, it is susceptible to SpO₂ artifacts and the challenges of accurately measuring respiratory rate at the bedside [58,59].
… Ranieri and colleagues have argued that incorporating such measures could help distinguish ARDS from other causes of hypoxemic respiratory failure and offer a more refined characterization of disease severity [39]. However, pulmonary permeability indices often rely on invasive sampling or proprietary biomarkers that are not available in routine clinical practice, limiting their feasibility outside research centers [39]. Similarly, the assessment of lung weight and aeration through advanced imaging techniques, such as computed tomography (CT) or magnetic resonance imaging (MRI), provides valuable insights into the extent of alveolar flooding and consolidation. Nevertheless, quantitative CT and MRI provide high-resolution data but are costly, expose patients to radiation (in the case of CT), and require patient transport, an impractical option for unstable ICU patients. MRI also demands lengthy scan times and specialized expertise. In addition, inter-institutional variability in imaging protocols and post-processing algorithms further hampers comparability and may introduce measurement error. Until these hurdles are overcome, the integration of advanced permeability and imaging parameters into routine ARDS definitions will remain primarily a research endeavor rather than a universal clinical standard [63].
C: 6. Conclusions. The main criticisms are the potential for overdiagnosis and variability in diagnostic tools, as well as the challenge of ensuring that the inclusion of new modalities truly improves diagnostic accuracy without leading to unnecessary treatments.
R: Accordingly, we have sharpened our focus on the risks of overdiagnosis and tool variability, emphasizing the need for context-specific validation of each new modality. We have also updated the text following the comments and suggestions of Reviewer 3. (Lines 391-405)
In summary, the development of ARDS diagnostic criteria and management strategies represent a significant effort to improve patient outcomes by adapting approaches to individual and contextual needs. Expanding the definition of ARDS to include non-invasive respiratory support tools, such as HFNC and NIV, recognizes their crucial role in modern clinical practice, particularly in resource-limited settings and during global health crises, like occurred for the COVID-19 pandemic. While these advancements undoubtedly improve diagnostic inclusivity and accessibility, they also bring challenges, such as potential overdiagnosis and variability in diagnostic tools like SpO₂/FiO₂ ratios and lung ultrasound. To avoid unnecessary treatments, it is essential that each new modality undergo rigorous validation to confirm that it truly enhances diagnostic accuracy and clinical outcomes. Future perspectives in ARDS research and management point towards integrating advanced physiological indices like biomarkers, imaging modalities, and personalized approaches to refine diagnosis and stratify disease severity.
Reviewer 2 Report
Comments and Suggestions for Authors
The manuscript provides a comprehensive and timely review of the evolving definitions and diagnostic criteria for Acute Respiratory Distress Syndrome (ARDS), particularly emphasizing the incorporation of non-invasive ventilation (NIV) and high-flow nasal cannula (HFNC) into diagnostic considerations. It is comprehensive, adding a brief paragraph on how POCUS training might be useful in expedient management would be helpful
Author Response
We thank the Reviewer very much, his/her comments, suggestions and feedback on our article allowed us to improve it.
C: The manuscript provides a comprehensive and timely review of the evolving definitions and diagnostic criteria for Acute Respiratory Distress Syndrome (ARDS), particularly emphasizing the incorporation of non-invasive ventilation (NIV) and high-flow nasal cannula (HFNC) into diagnostic considerations. It is comprehensive, adding a brief paragraph on how POCUS training might be useful in expedient management would be helpful
R: We thank the reviewer for these comments. We have added a small paragraph in “advantages” chapter on how POCUS training might be useful in ARDS management. (Lines 252-332)
Advantages and Practical Considerations
… Recent international guidelines have increasingly recognized the potential value of lung ultrasound (LUS), as introduced by the Kigali modification, although it has not yet been formally incorporated into the standard global definition of ARDS [11]. In particular to maximize the benefits of LUS in ARDS a structured point-of-care ultrasound (POCUS) training is essential. Studies have shown that clinicians can achieve basic competency in lung ultrasound after as few as 20 supervised examinations, leading to significant improvements in image acquisition, interpretation accuracy, and time to diagnosis [47–49]. By standardizing POCUS training, scanning protocols, and interpretation criteria, teams can reduce inter-operator variability and integrate LUS more reliably into early ARDS management pathways. A useful development is the LUS-ARDS score, a data driven and externally validated method based on LUS-scores from both the left and right lungs, combined with the identification of an abnormal pleural line in the antero-lateral regions [50]. Although more complex than the Kigali modification, it shows higher accuracy in diagnosing and excluding ARDS [51]. Another critical advantage of the new definition is its improved global applicability, including in resources-limited settings. This adaptability ensures that ARDS can be diagnosed and effectively managed even in low-income countries where access to advanced diagnostics and MV is often restricted [35,52] (Table 4).
Limitations
… By broadening the criteria to include non-intubated patients on HFNC or NIV, there is a risk of misclassifying other causes of acute hypoxemic respiratory failure as ARDS. For example non‐invasive support techniques may suffer some technical problems like mask leaks or malpositioning that may result in uneven PEEP delivery causing regional atelectasis and patchy aeration that appear as bilateral infiltrates. Similarly, modest fluid overload under insufficient positive pressure could generate interstitial B-lines indistinguishable from inflammatory edema. Under expanded HFNC/NIV-based criteria, these patients could be misclassified as ARDS, exposing patients to unnecessary treatments and resource utilization, potentially overburdening healthcare systems and diluting the specificity of ARDS as a distinct clinical entity [38]. In addition, the application of the new definition is also problematic in certain specific patient populations, such as those receiving V-V ECMO. Many V-V ECMO patients are awake and spontaneously breathing and may even be in ambient air (FiO₂ 21%). In such cases, traditional oxygenation-based diagnostic criteria such as PaO₂/FiO₂ or SpO₂/FiO₂ ratios can appear within normal limits despite the presence of severe lung pathology. As a result, these patients may not meet the formal ARDS criteria and could be erroneously excluded from diagnosis. As a result, the revised criteria may not fully address the diagnostic needs of ECMO-supported ARDS patients, leading to inconsistent classification [10]. …
… Additionally, the incorporation of LUS into ARDS diagnostics is valuable for bedside assessments, but on the other hand has some limitations, as its accuracy depends heavily on the clinician’s skill and experience and may be less effective in differentiating ARDS from other conditions with similar findings. In fact LUS findings such as B-lines, pleural line abnormalities, or subpleural consolidations, though common in ARDS, are not pathognomonic and may also be present in conditions like cardiogenic pulmonary edema or interstitial lung diseases. This overlap can complicate the differential diagnosis, especially in the absence of standardized interpretation criteria [39,51,55] (Table 4).
Reviewer 3 Report
Comments and Suggestions for Authors
-
Title & Abstract:
-
The title is informative and appropriate for a review article.
-
The abstract clearly outlines the scope, significance, and conclusions. Consider tightening language slightly to improve conciseness.
-
-
Lines 1–4 (Title and Author List):
-
Ensure the title uses consistent capitalization (“Old and New Definitions…”).
-
Confirm author affiliations and equal contribution notations are properly formatted per journal style.
-
-
Lines 15–42 (Abstract):
-
Excellent summary of ARDS evolution.
-
The sentence starting with “The Berlin definition…” (line 21) is dense; consider breaking it into two for clarity.
-
“Patients who worsen despite non-invasive ventilatory support” (line 24) could benefit from more clinical specificity.
-
Lines 36–40 introduce a nuanced proposal; consider clarifying what “advanced physiological or biomarker-based indicators” refers to with examples.
-
-
Lines 43–76 (Introduction):
-
Strong context-setting on the burden of LRTI and pneumonia.
-
Line 66: “indicate” should be “indicates” (subject-verb agreement).
-
Line 69: Consider rephrasing “recommended therapeutic measures” to specify what interventions are meant.
-
-
Lines 77–119 (Definition of ARDS):
-
Clear historical overview.
-
Lines 102–104: Good mention of the Kigali modification; it may help to include more on its limitations.
-
Line 118: Consider specifying examples of “personalized approach” (e.g., based on imaging, biomarkers, etc.).
-
-
Lines 121–127 (Table 1 & 2):
-
Ensure abbreviations are all defined at first mention.
-
Table 2 could benefit from brief notes or citations embedded to support ESICM recommendations.
-
-
Lines 130–158 (Management of ARDS):
-
Well-structured, but line 143’s long sentence may be broken for readability.
-
Line 156: Instead of "a more recent meta-analysis," cite the specific meta-analysis reference in-text.
-
-
Lines 159–182 (NIV and HFNC):
-
Comprehensive but could benefit from more structured subpoints (e.g., "Benefits," "Risks").
-
Line 171: Define P-SILI explicitly for readers unfamiliar with the term.
-
-
Lines 184–219 (New ARDS Criteria):
-
Clearly outlines evolution in diagnostic thresholds.
-
Table 3 is excellent but may benefit from color-coding or shading for distinction between criteria categories.
-
-
Lines 220–281 (Advantages and Limits):
-
Balanced analysis.
-
Lines 263–265: Consider rewording to emphasize that normal oxygenation under ECMO doesn’t preclude ongoing pathology.
-
Line 271: Add citation for "device variability" impacting SpO₂ accuracy.
-
Lines 290–319 (Future Perspectives):
-
Well-developed and insightful.
-
Lines 295–301: Define the correction factor and origin of the P/FP formula more explicitly.
-
Line 303: Explain the ROX index in greater detail and how it’s applied practically.
-
Lines 321–332 (Conclusions):
-
Strong summary.
-
Line 330: “integrating advanced physiological indices” could use one concrete example to strengthen clarity.
-
Lines 334–345 (Author Contributions & Conflicts):
-
Appropriate and clear. Ensure final submission matches journal formatting for contributions and correspondence.
-
References:
-
Comprehensive and current.
-
Double-check for formatting consistency (e.g., punctuation, DOI placement).
-
General Language & Structure:
-
Overall English is clear and professional.
-
Minor grammatical polishing (e.g., subject-verb agreement, sentence splitting) could enhance clarity.
-
Consider tightening some long sentences and avoiding redundancy across sections.
Author Response
We appreciated very much the detailed suggestions and comments of the Reviewer. Thanks to them we had the opportunity to significantly improve the paper.
C: Title & Abstract:
- The title is informative and appropriate for a review article.
- The abstract clearly outlines the scope, significance, and conclusions. Consider tightening language slightly to improve conciseness.
R: We thank the Reviewer for the comment. In the revised version of the paper we have used a clearer segmentation of the Abstract to improve readability and comprehension. Actually, it was also the change we made according with the comments of Reviewer 1. (Lines 16-43)
Low respiratory tract infections remain a leading cause of morbidity and mortality among Intensive Care Unit patients, with severe cases often progressing to acute respiratory distress syndrome (ARDS). This life-threatening syndrome results from alveolar–capillary membrane injury, causing refractory hypoxemia and respiratory failure. …
The 2012 Berlin definition standardized ARDS diagnosis but excluded patients on non-invasive ventilation (NIV) or high-flow nasal cannula (HFNC) modalities, which are increasingly used, especially after the COVID-19 pandemic. By excluding these patients, diagnostic delays can occur, risking progression of lung injury despite ongoing support. Indeed, sustained, vigorous respiratory efforts under non‑invasive modalities carry a significant potential for patient self‑inflicted lung injury (P‑SILI), underscoring the need to broaden diagnostic criteria to encompass these increasingly common therapies.
Recent proposals expand ARDS criteria to include NIV and HFNC, lung ultrasound, and the SpO₂/FiO₂ ratio adaptations designed to improve diagnosis in resource-limited settings lacking arterial blood gases or advanced imaging.
However, broader criteria risk overdiagnosis and create challenges in distinguishing ARDS from other causes of acute hypoxemic failure. …
To overcome these limitations, a more nuanced diagnostic framework is needed, one that incorporates individualized therapeutic strategies, emphasizes lung-protective ventilation, and integrates advanced physiological or biomarker-based indicators like IL-6, IL-8, IFN-γ associated with worse outcomes.
Such an approach has the potential to improve patient stratification, enable more targeted interventions, and ultimately support the design and conduct of more effective interventional studies.
C: Lines 1–4 (Title and Author List):
- Ensure the title uses consistent capitalization (“Old and New Definitions…”).
- Confirm author affiliations and equal contribution notations are properly formatted per journal style.
R: Accordingly, we corrected the title using consistent capitalization; also, we reviewed and adjusted the author names, affiliations and equal contribution symbols to align with the Journal’s formatting guidelines. (Lines 2-15)
Old and New Definitions of Acute Respiratory Distress Syndrome (ARDS): An Overview of Practical Considerations and Clinical Implications
Cesare Biuzzi1, †, Elena Modica1, †, Noemi De Filippis1, Daria Pizzirani1, Benedetta Galgani1, Agnese Di Chiaro1, Daniele Marianello2, Federico Franchi2, Fabio Silvio Taccone3, Sabino Scolletta1, *
1 Department of Medical Science, Surgery and Neurosciences, Urgency-Emergency Anesthesia and Intensive Care Unit, University Hospital of Siena, 53100 Siena, Italy.
2 Department of Medical Science, Surgery and Neurosciences, Cardiothoracic and Vascular Anesthesia and Intensive Care Unit, University Hospital of Siena, 53100 Siena, Italy.
3 Department of Intensive Care, Hôpital Universitaire de Bruxelles (HUB), Université Libre de Bruxelles (ULB), Brussels, Belgium.
† These authors contributed equally to this work
* Correspondence: sabino.scolletta@unisi.it
C: Lines 15–42 (Abstract):
- Excellent summary of ARDS evolution.
- The sentence starting with “The Berlin definition…” (line 21) is dense; consider breaking it into two for clarity.
- “Patients who worsen despite non-invasive ventilatory support” (line 24) could benefit from more clinical specificity.
- Lines 36–40 introduce a nuanced proposal; consider clarifying what “advanced physiological or biomarker-based indicators” refers to with examples.
R: We agree with the Reviewer. We have changed the text accordingly. Actually, it was also the change we made according with the comments of Reviewer 1. (Lines 16-43)
Low respiratory tract infections remain a leading cause of morbidity and mortality among Intensive Care Unit patients, with severe cases often progressing to acute respiratory distress syndrome (ARDS). This life-threatening syndrome results from alveolar–capillary membrane injury, causing refractory hypoxemia and respiratory failure. …
The 2012 Berlin definition standardized ARDS diagnosis but excluded patients on non-invasive ventilation (NIV) or high-flow nasal cannula (HFNC) modalities, which are increasingly used, especially after the COVID-19 pandemic. By excluding these patients, diagnostic delays can occur, risking progression of lung injury despite ongoing support. Indeed, sustained, vigorous respiratory efforts under non‑invasive modalities carry a significant potential for patient self‑inflicted lung injury (P‑SILI), underscoring the need to broaden diagnostic criteria to encompass these increasingly common therapies.
Recent proposals expand ARDS criteria to include NIV and HFNC, lung ultrasound, and the SpO₂/FiO₂ ratio adaptations designed to improve diagnosis in resource-limited settings lacking arterial blood gases or advanced imaging.
However, broader criteria risk overdiagnosis and create challenges in distinguishing ARDS from other causes of acute hypoxemic failure. …
To overcome these limitations, a more nuanced diagnostic framework is needed, one that incorporates individualized therapeutic strategies, emphasizes lung-protective ventilation, and integrates advanced physiological or biomarker-based indicators like IL-6, IL-8, IFN-γ associated with worse outcomes.
Such an approach has the potential to improve patient stratification, enable more targeted interventions, and ultimately support the design and conduct of more effective interventional studies.
C: Lines 43–76 (Introduction):
- Strong context-setting on the burden of LRTI and pneumonia.
- Line 66: “indicate” should be “indicates” (subject-verb agreement).
- Line 69: Consider rephrasing “recommended therapeutic measures” to specify what interventions are meant.
R: As requested, we have changed the text of the paper. In addition, the Reviewer 1 suggested some similar changes. (Lines 46-84)
… The need for respiratory support is one of the primary reasons for ICU admission in these patients with treatment options ranging from low- and high-flow oxygen therapy to non-invasive ventilation (NIV) and invasive mechanical ventilation [3]. … … For this reasons, these patients require comprehensive monitoring, which includes imaging studies such as lung ultrasonography (LUS), chest X-rays, CT-scans as well as continuous assessment of respiratory drive, respiratory effort, and lung-distending pressures in order to prevent Ventilator-Induced Lung Injury (VILI) and Patient Self-Inflicted Lung Injury (P-SILI) particularly in those receiving NIV and HFNC support where monitoring can be less rigorous [5,6]. …
Indeed, one recent international observational study involving several intensive care units across 50 countries indicates that ARDS remains underrecognized, with clinician recognition rates as low as 60%, particularly for mild cases; this phenomenon is associated with insufficient application of recommended therapeutic measures (such as low tidal volume ventilation, appropriate use of positive end-expiratory pressure etc., Table 2) which contribute to persistently high mortality rates (approximately 40%) [8]. …
C: Lines 77–119 (Definition of ARDS):
- Clear historical overview.
- Lines 102–104: Good mention of the Kigali modification; it may help to include more on its limitations.
- Line 118: Consider specifying examples of “personalized approach” (e.g., based on imaging, biomarkers, etc.).
R: We included Kigali limitations and specified examples of “personalized approach”. We have updated the text taking into consideration also some similar suggestions of Reviewer 1. (Lines 85-128)
… First described in 1967 [12] his diagnostic criteria were revised in 1988 [13], and evolved significantly over the decades since 1994 when the American-European Consensus Conference (AECC) formalized diagnostic criteria [14] with the goal of refining and unifying the various existing definitions of ARDS and Acute Lung Injury (ALI). In particular AECC established diagnostic criteria for ALI as a syndrome with acute onset, …
These limitations prompted the development of the Berlin definition in 2012 [9] whose key diagnostic criteria include an acute onset occurring within one week of a recognized clinical event or a significant worsening of respiratory symptoms; …. This definition was primarily focused on patients undergoing invasive mechanical ventilation, effectively excluding those treated with non-invasive respiratory support. To overcome these limitations, several adaptations have been proposed. The Kigali modification (2016) substituted the PaO₂/FiO₂ ratio with the SpO₂/FiO₂ ratio, permitted lung ultrasound as an alternative to chest X-rays, and removed the PEEP requirement [18]. However, these adaptations introduce their own challenges. Lung ultrasound is highly operator-dependent, with image quality and interpretation varying by examiner expertise, while SpO₂/FiO₂ ratios can be skewed by motion artifacts, poor perfusion, and probe positioning [19,20]. In 2023, the European Society of Intensive Care Medicine (ESICM) updated its guidelines on ARDS, revising key aspects related to its definition, phenotyping, and respiratory support strategies (Table 1 and 2, Figure 1) [11]. Finally, a global consensus (2023–2024) advocated for expanding the ARDS definition to include patients treated with HFNC (≥30 L/min) and NIV with equivalent PEEP, supported the use of SpO₂/FiO₂ for oxygenation assessment, and encouraged bedside diagnostic tools such as lung ultrasound (LUS). These updates aim to broaden diagnostic inclusivity, especially in low-resource settings, facilitate earlier recognition and intervention, and support a more personalized approach (e.g., combining imaging with biomarkers etc.) to patient management based on disease severity and the type of ventilatory support. However, challenges persist in avoiding overdiagnosis, maintaining diagnostic consistency across different clinical settings, and improving the accuracy and reliability of tools used to assess oxygenation [20].
C: Lines 121–127 (Table 1 & 2):
- Ensure abbreviations are all defined at first mention.
- Table 2 could benefit from brief notes or citations embedded to support ESICM recommendations.
R: Thanks for these suggestions. We added the complete definition for abbreviations and added brief notes or citations embedded to support ESICM recommendations. (Lines 130-138)
Table 1: “ARDS: Acute Respiratory Distress Syndrome”
Note Table 2: Recommendations based on ESICM 2023 consensus [11]; see also supporting literature [12–16] for specific interventions.
Lines 130–158 (Management of ARDS):
- Well-structured, but line 143’s long sentence may be broken for readability.
- Line 156: Instead of "a more recent meta-analysis," cite the specific meta-analysis reference in-text.
R: We changed the text, as requested. (Lines 153-178)
The management of ARDS has advanced significantly in recent years. … Furthermore, prone positioning enhances ventilation–perfusion matching and reduces alveolar collapse in dorsal regions. It has proven one of the most effective strategies for managing severe ARDS (PaO₂/FiO₂ below 150 mm Hg) significantly reducing mortality in this high-risk group [8,23–25]. … While the EOLIA trial did not demonstrate a significant reduction in 60-day mortality in ECMO-treated patients compared to those receiving only invasive mechanical ventilation, it highlighted an increased risk of bleeding or ischemic stroke. However, a meta-analysis by Munshi et al. (2019), which included both the CESAR and EOLIA trials, demonstrated a significant reduction in 60-day mortality in patients treated with V-V ECMO [26,28,29].
C: Lines 159–182 (NIV and HFNC):
- Comprehensive but could benefit from more structured subpoints (e.g., "Benefits," "Risks").
- Line 171: Define P-SILI explicitly for readers unfamiliar with the term.
R: We have changed the text accordingly. (Lines 180-213)
…
Benefits
Noninvasive ventilation (NIV) and high-flow nasal cannula (HFNC) offer early respiratory support in patients with mild-to-moderate ARDS. These non-invasive strategies offer several advantages: in particular, HFNC has become a critical respiratory support option during the COVID‑19 pandemic, thanks to its ability to deliver heated, humidified oxygen at high flow rates. This reduces respiratory effort and improves oxygenation [26,29–31]. Moreover, they aim to reduce the need for invasive MV, thereby avoiding complications such as ventilator-induced lung injury (VILI), sedation-related risks, and ventilator-associated pneumonia [32]. Clinical trials have shown that HFNC can lower tracheal intubation rates and improve survival compared to conventional oxygen therapy [33].
Risks
Despite these advantages, NIV and HFNC carry significant hazards. Vigorous respiratory efforts can exacerbate lung injury through patient-self inflicted lung injury (P-SILI) - a life‑threatening condition in which excessive respiratory effort and forceful inspiration exacerbate lung damage, particularly in patients with preexisting lung injury, severe ARDS, or a high inspiratory drive [34]. Furthermore, delayed intubation in patients who fail NIV or HFNC has been associated with increased mortality, highlighting the importance of close monitoring and timely escalation to invasive MV when necessary [35,36]. …
C: Lines 184–219 (New ARDS Criteria):
- Clearly outlines evolution in diagnostic thresholds.
- Table 3 is excellent but may benefit from color-coding or shading for distinction between criteria categories.
R: Thanks for these comments. We have modified the article, as requested. For the table modifications, please refer to the attached file. (Lines 249-251)
C: Lines 220–281 (Advantages and Limits):
- Balanced analysis.
- Lines 263–265: Consider rewording to emphasize that normal oxygenation under ECMO doesn’t preclude ongoing pathology.
- Line 271: Add citation for "device variability" impacting SpO₂ accuracy.
R: We thank the Reviewer for the comment. We changed and improved the paper, as requested. (Lines 252-332)
Advantages and Practical Considerations
… Recent international guidelines have increasingly recognized the potential value of lung ultrasound (LUS), as introduced by the Kigali modification, although it has not yet been formally incorporated into the standard global definition of ARDS [11]. In particular to maximize the benefits of LUS in ARDS a structured point-of-care ultrasound (POCUS) training is essential. Studies have shown that clinicians can achieve basic competency in lung ultrasound after as few as 20 supervised examinations, leading to significant improvements in image acquisition, interpretation accuracy, and time to diagnosis [47–49]. By standardizing POCUS training, scanning protocols, and interpretation criteria, teams can reduce inter-operator variability and integrate LUS more reliably into early ARDS management pathways. A useful development is the LUS-ARDS score, a data driven and externally validated method based on LUS-scores from both the left and right lungs, combined with the identification of an abnormal pleural line in the antero-lateral regions [50]. Although more complex than the Kigali modification, it shows higher accuracy in diagnosing and excluding ARDS [51]. … This adaptability ensures that ARDS can be diagnosed and effectively managed even in low-income countries where access to advanced diagnostics and MV is often restricted [35,52] (Table 4).
Limitations
… By broadening the criteria to include non-intubated patients on HFNC or NIV, there is a risk of misclassifying other causes of acute hypoxemic respiratory failure as ARDS. For example non‐invasive support techniques may suffer some technical problems like mask leaks or malpositioning that may result in uneven PEEP delivery causing regional atelectasis and patchy aeration that appear as bilateral infiltrates. Similarly, modest fluid overload under insufficient positive pressure could generate interstitial B-lines indistinguishable from inflammatory edema. Under expanded HFNC/NIV-based criteria, these patients could be misclassified as ARDS, exposing patients to unnecessary treatments and resource utilization, potentially overburdening healthcare systems and diluting the specificity of ARDS as a distinct clinical entity [38]. In addition, the application of the new definition is also problematic in certain specific patient populations, such as those receiving V-V ECMO. Many V-V ECMO patients are awake and spontaneously breathing and may even be in ambient air (FiO₂ 21%). In such cases, traditional oxygenation-based diagnostic criteria such as PaO₂/FiO₂ or SpO₂/FiO₂ ratios can appear within normal limits despite the presence of severe lung pathology. As a result, these patients may not meet the formal ARDS criteria and could be erroneously excluded from diagnosis. As a result, the revised criteria may not fully address the diagnostic needs of ECMO-supported ARDS patients, leading to inconsistent classification [10].
… Additionally, the incorporation of LUS into ARDS diagnostics is valuable for bedside assessments, but on the other hand has some limitations, as its accuracy depends heavily on the clinician’s skill and experience and may be less effective in differentiating ARDS from other conditions with similar findings. In fact LUS findings such as B-lines, pleural line abnormalities, or subpleural consolidations, though common in ARDS, are not pathognomonic and may also be present in conditions like cardiogenic pulmonary edema or interstitial lung diseases. This overlap can complicate the differential diagnosis, especially in the absence of standardized interpretation criteria [39,51,55] (Table 4).
C: Lines 290–319 (Future Perspectives):
- Well-developed and insightful.
- Lines 295–301: Define the correction factor and origin of the P/FP formula more explicitly.
- Line 303: Explain the ROX index in greater detail and how it’s applied practically.
R: We changed the text as requested, also accordingly to some similar suggestions of Reviewer 1. (Lines 335-390)
The ongoing evolution of ARDS diagnostic criteria reflects the need for greater precision in identifying and managing this complex syndrome. However, several criticisms have emerged: variability and potential inaccuracies introduced by new indices and diagnostic tools; limited feasibility and lack of standardization of advanced physiological and imaging parameters; risks of overdiagnosis and consequent delays in appropriate interventions; and difficulties in implementing these approaches across diverse healthcare settings. Various authors have proposed refinements to improve diagnostic accuracy and to adapt the criteria more effectively for both clinical practice and research purposes [56].
… P/FP is calculated by dividing the traditional PaO₂/FiO₂ ratio by the applied PEEP and then multiplying the result by a correction factor of 10. This correction factor was introduced to keep the P/FP ratio on a scale comparable to the traditional PaO₂/FiO₂ ratio, making it easier to interpret clinically. The same concept can be used for S/FP calculation. This formula adjusts for the contribution of PEEP to oxygenation, reflecting the interaction between alveolar recruitment and gas exchange efficiency [57]. However, several sources of inaccuracy may limit their reliability in fact they are vulnerable to practice‐dependent PEEP settings and the arbitrary “×10” correction factor can reclassify patients without improving prediction. In addition, in noninvasive support (e.g., HFNC), true PEEP is neither standardized nor directly measured, and SpO₂ is prone to artifacts (poor perfusion, pigmentation), risking misclassification of oxygenation status [57].
Some authors used the ROX index calculated as SpO₂/FiO₂ ratio/respiratory rate as an indicator of the severity of respiratory failure in patients with ARDS receiving non-invasive ventilation. A higher ROX index suggests better oxygenation efficiency relative to breathing effort, helping clinicians identify patients likely to benefit from continued non-invasive support versus those needing intubation. Despite its simplicity, the ROX index’s utility in ARDS is limited. It was originally developed and validated in pneumonia patients on high-flow nasal cannula, so its performance in a broader ARDS population or in those receiving NIV may differ. Moreover, it is susceptible to SpO₂ artifacts and the challenges of accurately measuring respiratory rate at the bedside [58,59].
… Ranieri and colleagues have argued that incorporating such measures could help distinguish ARDS from other causes of hypoxemic respiratory failure and offer a more refined characterization of disease severity [39]. However, pulmonary permeability indices often rely on invasive sampling or proprietary biomarkers that are not available in routine clinical practice, limiting their feasibility outside research centers [39]. Similarly, the assessment of lung weight and aeration through advanced imaging techniques, such as computed tomography (CT) or magnetic resonance imaging (MRI), provides valuable insights into the extent of alveolar flooding and consolidation. Nevertheless, quantitative CT and MRI provide high-resolution data but are costly, expose patients to radiation (in the case of CT), and require patient transport, an impractical option for unstable ICU patients. MRI also demands lengthy scan times and specialized expertise. In addition, inter-institutional variability in imaging protocols and post-processing algorithms further hampers comparability and may introduce measurement error. Until these hurdles are overcome, the integration of advanced permeability and imaging parameters into routine ARDS definitions will remain primarily a research endeavor rather than a universal clinical standard [63].
C: Lines 321–332 (Conclusions):
- Strong summary.
- Line 330: “integrating advanced physiological indices” could use one concrete example to strengthen clarity.
R: Thanks for the comment. The text was changed, as requested. (Lines 391-405)
In summary, the development of ARDS diagnostic criteria and management strategies represent a significant effort to improve patient outcomes by adapting approaches to individual and contextual needs. Expanding the definition of ARDS to include non-invasive respiratory support tools, such as HFNC and NIV, recognizes their crucial role in modern clinical practice, particularly in resource-limited settings and during global health crises, like occurred for the COVID-19 pandemic. While these advancements undoubtedly improve diagnostic inclusivity and accessibility, they also bring challenges, such as potential overdiagnosis and variability in diagnostic tools like SpO₂/FiO₂ ratios and lung ultrasound. To avoid unnecessary treatments, it is essential that each new modality undergo rigorous validation to confirm that it truly enhances diagnostic accuracy and clinical outcomes. Future perspectives in ARDS research and management point towards integrating advanced physiological indices like biomarkers, imaging modalities, and personalized approaches to refine diagnosis and stratify disease severity.
C: Lines 334–345 (Author Contributions & Conflicts):
- Appropriate and clear. Ensure final submission matches journal formatting for contributions and correspondence.
R: We made some adjustments to contributions and correspondence to ensure consistency and uniformity in style. (Lines 406-417)
Author Contributions: C.B. and E.M. contributed equally to this study. C.B. and E.M. have given substantial contributions to the conception, writing of the manuscript, acquisition, analysis, and interpretation of the data. N.D.F contributed to drafting the introduction section. C.B. and E.M. realized the tables. F.S.T. and S.S. created the figure. A.D.C., D.P., B.G. and D.M. revised it critically. S.S., F.F and F.S.T. supervised and revised it critically. All authors read and approved the final version of the manuscript. ...
C: References:
- Comprehensive and current.
- Double-check for formatting consistency (e.g., punctuation, DOI placement).
R: We made some adjustments to the references to ensure consistency and uniformity in style.
C: General Language & Structure:
- Overall English is clear and professional.
- Minor grammatical polishing (e.g., subject-verb agreement, sentence splitting) could enhance clarity.
- Consider tightening some long sentences and avoiding redundancy across sections.
R: Thanks for the valuable feedback. We have carefully reviewed the manuscript and made some minor grammatical corrections, improved sentence structure for clarity, and reduced redundancy where appropriate to enhance overall readability.
Round 2
Reviewer 1 Report
Comments and Suggestions for Authors
The manuscript has been improved, as requested. No further comments.